# Short-Term RCT of Increased Dietary Potassium from Potato or Potassium Gluconate: Effect on Blood Pressure, Microcirculation, and Potassium and Sodium Retention in Pre-Hypertensive-to-Hypertensive Adults

**DOI:** 10.3390/nu13051610

**Published:** 2021-05-11

**Authors:** Michael S. Stone, Berdine R. Martin, Connie M. Weaver

**Affiliations:** Nutrition Science, Purdue University, West Lafayette, IN 47907, USA; mstone@sdsu.edu (M.S.S.); bmartin1@purdue.edu (B.R.M.)

**Keywords:** potassium, sodium, blood pressure, controlled feeding study, microcirculation, retention, cardiometabolic risk

## Abstract

Increased potassium intake has been linked to improvements in cardiovascular and other health outcomes. We assessed increasing potassium intake through food or supplements as part of a controlled diet on blood pressure (BP), microcirculation (endothelial function), and potassium and sodium retention in thirty pre-hypertensive-to-hypertensive men and women. Participants were randomly assigned to a sequence of four 17 day dietary potassium treatments: a basal diet (control) of 60 mmol/d and three phases of 85 mmol/d added as potatoes, French fries, or a potassium gluconate supplement. Blood pressure was measured by manual auscultation, cutaneous microvascular and endothelial function by thermal hyperemia, utilizing laser Doppler flowmetry, and mineral retention by metabolic balance. There were no significant differences among treatments for end-of-treatment BP, change in BP over time, or endothelial function using a mixed-model ANOVA. However, there was a greater change in systolic blood pressure (SBP) over time by feeding baked/boiled potatoes compared with control (−6.0 mmHg vs. −2.6 mmHg; *p* = 0.011) using contrast analysis. Potassium retention was highest with supplements. Individuals with a higher cardiometabolic risk may benefit by increasing potassium intake. This trial was registered at ClinicalTrials.gov as NCT02697708.

## 1. Introduction

Potassium has recently been listed as an essential nutrient of concern according to the Dietary Guidelines Advisory Committee for Americans [1,2,3]. Increases in potassium intake have been linked to improvements in cardiovascular and other metabolic health outcomes. Blood pressure (BP) is currently the primary health criterion for evaluating potassium requirements. Cardiovascular disease (CVD) is responsible for the majority of deaths worldwide [4]. Hypertension, or high blood pressure (BP), the primary risk factor for cardiovascular disease (CVD) and other circulatory diseases, is a leading cause of mortality worldwide [5]. Findings from the Agency for Healthcare Research and Quality (AHRQ) report, which informed the recently released 2019 Dietary Reference Intakes (DRIs) for potassium, concluded, with a moderate strength of evidence, that increasing potassium intake decreases BP, particularly in those with hypertension [6]. However, only 4 of the 18 randomized controlled trials assessed by the AHRQ were dietary interventions, the rest involved potassium supplementation. Numerous meta-analyses conducted over the past 30 years support the AHRQ report, finding a positive relationship between potassium supplementation and BP reduction in pre-hypertensive-to-hypertensive adults [7,8,9,10,11,12]. In contrast, overall findings on the effect of increased dietary potassium intake and BP have been conflicting, with some studies confirming benefit [13] and others finding null results [14,15]. The majority of these trials have utilized dietary advice or coaching to increase participant potassium consumption. This lack of control is a major limitation and may explain differing results. Consequently, the DRI panel concluded that there was insufficient evidence to determine the potassium intake for Chronic Disease Risk Reduction (CDRR) or for an Estimated Average Requirement (EAR). Therefore, the Adequate Intake (AI) of potassium based on the highest medium intakes was set for men at ~85 mmol/d (~3400 mg/d) and women at ~65 mmol/d (~2600 mg/d).

Evidence on the effect of increased dietary potassium on BP from clinical trials is extremely limited, and there are currently no known controlled feeding interventions investigating potassium as the primary variable of interest. Because of this, controlled feeding trials utilizing a dietary pattern that increases potassium consumption have been used to infer effects of potassium intake on BP. These findings have primarily been extrapolated from the Dietary Approaches to Stop Hypertension (DASH) studies [16,17,18]. The DASH intervention showed that a dietary pattern focused on increased fruit and vegetable intake, fiber, and low-fat dairy products, with reduced intakes in fat and sodium (Na) improves BP outcomes compared to the average American diet [16]. Although the initial DASH trial diet did lead to increases in dietary potassium (DASH diet = 105 mmol/d or 4100 mg/d, DASH combination diet (DASH diet + low-fat dairy, low saturated and total fat) = 113 mmol/d or 4400 mg/d) and BP reduction, these outcomes cannot be attributed to potassium alone due to the other concurrent nutrient modifications [19].

The effect of dietary potassium intake on BP, as well as other health outcomes, may be related to overall potassium status. A specific biomarker of potassium status has yet to be clearly identified. Potassium retention from dose–response studies is useful to establish potassium adequacy. However, potassium balance studies are limited in number due to the necessary labor and time associated with such interventions for both participants and researchers. Only seven studies with assessment of complete intake, urinary, and fecal losses were available to the DRI committee. These studies showed an increase in urinary potassium with increasing intake (primary mode of excretion) [20], fecal losses averaging approximately 10.7 mmol/d (420 mg/d)) at intakes ranging from 59 to 100 mg/d (2300 to 3900 mg/d) [21,22,23], as well as variable losses due to sweat [24,25,26,27,28]. Given the limited number and overall heterogeneity in study design and findings, the committee concluded that there was insufficient evidence to establish potassium intake recommendations.

Potassium and sodium (Na) have a dynamic physiological relationship and the way in which these nutrients interact to influence fluid balance has important health implications, primarily related to BP and CVD risk. Studies have shown that consuming potassium salts in place of Na salts increases urinary Na excretion [29] and decreases CVD-related mortality risk [30]. Trials also show improvements in BP with higher potassium intakes in the context of a high-Na diet [31], while others show no improvement over a Na intake that is low [13,32]. Understanding the role of potassium retention and how it relates to Na retention, and ultimately potassium adequacy, is especially important in at-risk populations such as pre-hypertensives/hypertensives, whose adequacy levels may be different.

The goal of this research was to utilize a controlled feeding study to examine the effect of increased dietary potassium from different sources (potato sources and a potassium supplement) on BP and microvascular outcomes and potassium and sodium retention compared with a control diet of usual potassium intake. We hypothesized that the interventions with an increased potassium intake would reduce BP, improve microvascular function, increase potassium retention, and decrease Na retention compared with the control diet.

## 2. Methods

### 2.1. Subjects

Subjects (N = 30, age 21 years and older) were recruited through a variety of different techniques at Purdue University and the surrounding area. Subjects were pre-hypertensive-to-hypertensive (SBP > 120 mmHg) men and women who were otherwise apparently healthy. Exclusion criteria included medication to treat hyper or hypotension (hypertensive monotherapy was allowed later in recruitment), history of myocardial infarction, diabetes mellitus, renal disease, GI disease, pancreatitis, cholestatic liver disease, cancer, smoking, use of illegal drugs, or excessive alcohol intake, subjects who were pregnant or lactating, allergy or intolerance to study foods, unwillingness to refrain from dietary supplements, and weight loss > 3 kg in the past 2 months. Screening consisted of two visits at least one week apart. Subjects came to the Purdue Nutrition Science clinical suite, where their consent was taken, had their BP measured, gave a blood draw, and filled out screening questionnaires. The second screening visit consisted of another BP measurement to ensure eligibility. When eligible, subjects completed a 4 day run-in period, and continuing subjects were randomized to a computer-generated allocation to one of 24 possible sequences of the 4 phases: control, potato, French fry (FF), and potassium (K) gluconate interventions. Subjects were enrolled from May 2016 to September 2017. All participants gave their written, informed consent for inclusion before they participated in this study. This study was conducted in accordance with the Declaration of Helsinki and the study protocol was approved by the Purdue University Institutional Review Board (Protocol #1511016780).

### 2.2. Dietary Intervention

The study design was a randomized, cross-over, controlled feeding trial, with the primary outcomes of end-of-treatment SBP and DBP and potassium and sodium retention and secondary outcomes of change in SBP and DBP over each intervention, as well as changes in microvascular function. After initial screening, eligible subjects entered the 4 day run-in period, designed to access participants’ ability to adhere to all study measures. During the run-in, subjects came to the clinical facility each morning, had their BP taken, either stayed to eat breakfast or had all their meals and snacks for the day packed out in a cooler to take with them. After the run-in, participants were assigned to a random order of four 16 day dietary potassium interventions including a basal diet (control) of ~60 mmol/d (2300 mg potassium/d) and three phases of an additional 26 mmol/d (1000 mg potassium/d) for a total potassium intake of ~85 mmol/d (3300 mg/d) in the form of potatoes (baked, boiled, or pan-heated with no additional fat), French fries (FF), or a K gluconate supplement. Each intervention was separated by two or more weeks of wash out. A 4 day menu cycle at three calorie levels (1800, 2200, and 2600 kcal/d) assigned for weight maintenance was designed to manipulate potassium levels while keeping other micro- (e.g., Na, Ca), macro- (e.g., fat, protein, carbohydrate), and other (e.g., fiber) nutrients constant. Subjects picked up study foods on days 1, 4, 6, 8, 11, 13, 15, and 16. For each day of controlled feeding, the subjects recorded their intake by checking off each item listed on the daily menu sheets provided. They were also instructed to indicate whether they ate any non-study foods, and/or whether they did not eat all the study foods that they were provided. The food and beverages were prepared with deionized water and weighed to the nearest one-tenth gram on digital scales. Meal composites of each menu cycle day were homogenized and chemically analyzed. Potassium gluconate supplements were sent out to a food chemistry lab for analysis (Eurofins Food Chemistry Testing Inc., Madison, WI, USA) Energy intakes were adjusted as necessary with snack items that contained mostly sugar. Participants were instructed to return any uneaten items to access compliance.

### 2.3. Blood Pressure and Microvascular Measures

A slightly modified version of the DASH BP assessment protocol was utilized to measure BP [16,33]. Blood pressure was measured by trained and certified study staff using manual auscultation with a McKesson Deluxe Aneroid Sphygmomanometer (Richmond, VA, USA), after the participant sat for at least 5 min at rest in a quiet, temperature-controlled room. Three readings (first and fifth Korotkoffs sounds) were taken from the left arm at each visit and averaged. Measurements were taken in the morning, after an overnight fast, at approximately the same time. We assessed BP on days 1, 4, 6, 8, 11, 13, 15, 16, and 17 during each phase. Results for differences in both systolic blood pressure (SBP) and diastolic blood pressure (DBP) were assessed at the end of treatment (average of day 15, 16, and 17 measurements). Secondary BP analysis assessed change in BP over time (end of treatment—baseline), with baseline BP defined as an average of measures from days 1 and 4 of each intervention period.

Cutaneous microvascular and endothelial function were assessed via thermal hyperemia, utilizing laser Doppler flowmetry (LDF), at baseline (day 1 or 2 of first intervention) and at the end of each intervention (day 15 or 16). Localized heating of the skin is a reliable method to measure nitric oxide (NO)-dependent dilation, and assess microvascular function in different vascular pathologies [34]. Cutaneous red blood cell flux was measured with an integrated LDF probe placed in a local heating unit (Skin Heater/Temperature Monitor SHO2, Moor Instruments, Devon, UK) positioned medially on the left forearm. Baseline flux measurements were collected at a temperature of 33 °C for 15 min, after which the heating probe was increased to 42 °C at a rate of 0.5 °C every 5 s for 45 min (to achieve a 10 min plateau), and finally maximal vasodilation was induced by increasing skin temperature to 43 °C for approximately 20 min [34,35]. Mean arterial pressure (MAP) was measured on the right arm throughout the protocol (every ~5 min) using an automated blood pressure monitor (Omron BP791IT; Omron Global, Hoffman Estates IL, USA). Cutaneous vascular conductance (CVC) was determined as flux divided by MAP, expressed as a percent of maximal vasodilation of the local site (% CVC max) [35].

### 2.4. Mineral Balance and Net Absorption

Methods for assessing mineral excretion and retention were similar to previous works [23,36]. Complete daily urine and feces were collected in acid-washed containers for each 16 day intervention of this study. Instructions and all necessary supplies (e.g., containers for urine and stool collections) were provided. Urine was pooled as 24 h collections, and analyzed for creatinine to assess collection compliance. Fecal samples were also pooled for each 24 h period and polyethylene glycol (PEG) recovery (turbidimetric assay), administered as two 500 mg capsules/three times per day, and were used to assess compliance. For mineral analysis, urine was acidified with 1% (by vol) HCL and stored at −20 °C. Fecal samples were homogenized with ultra-pure water and concentrated HCl using a laboratory stomacher (Tekmar Co, Cincinnati, OH, USA), heated in a drying oven at 50 °C for approximately 24 h, reduced to ash in a muffle furnace at 600 °C for 96 h, and diluted in 1 N HNO_3_ for mineral analysis. Urine was analyzed for creatinine, potassium and Na. Stools were analyzed for PEG and minerals. Dietary, urinary, and fecal minerals were measured by ICP-OES (5100 PC; Perkins Elmer, Waltham, MA, USA). Unacidified urine was measured for creatinine by a kinetic modification of Jaffe’s colorimetric assay (Cobas Mira Plus; Roche Diagnostic Systems, Nutley, NJ, USA). The first 2 days of each phase were excluded from balance calculations as equilibration days, as was the last day (day 16), which was an in-clinic testing day. Thus, a 13 day period was assessed for potassium and Na excretion and retention for each intervention. Balance and percent absorption (%ab) were determined using the following equations averaged over the 13 days:

Daily potassium/sodium balance (mg/d) = daily potassium/sodium intake (mg/d) − daily potassium/sodium excretion (mg/d) (urine and stools).

Percent (%) Net absorption = daily potassium/sodium intake (mg/d) − daily potassium/sodium fecal excretion (mg/d) × 100.

#### Statistical Methods

The means ± SE were calculated for all outcome measures. Differences in SBP, DBP, endothelial function (measured as percent of cutaneous vascular conductance max (% CVC max), and potassium and sodium excretion and retention (absolute and % of intake) among treatments were analyzed using a mixed-model ANOVA with Tukey post hoc analyses (*p* < 0.05). The primary BP outcome was mean differences among treatments in end-of-treatment BP (average of day 15, 16, 17), with adjustment for baseline BP (average of day 1 and 4) as a covariate (*p* < 0.0001). Secondary BP outcomes were change in BP over time (end of treatment—baseline) using a mixed-model ANOVA and contrast analysis comparing the potato intervention to the control. A sample size of 30 cross-over participants with two-sided alpha = 0.05 provided 80% power to detect differences in the primary outcome of differences in SBP of 3.8 mmHg at the end of treatment, and a sample size of 20 participants were needed to detect differences in potassium and Na retention of 9 mmol/d (355 mg/d) and 19 mmol/d (441 mg/d) or 10 and 14% of daily intake, respectively. All computations were performed using JMP (SAS Institute, Cary, NC, USA) statistical software. Alpha was set at 0.05. All results reported as the means ± SE unless otherwise noted.

## 3. Results

### 3.1. Baseline Characteristics of Study Subjects

Data from 30 subjects were included in the final analyses (Table 1). On average, subjects were 48.2 ± 15.0 y, had a BMI bordering overweight to obese (31.4 ± 6.1 kg/m^2^), and SBP and DBP of 133.6 ± 12.2 mmHg and 85.5 ± 8.6 mmHg, respectively (all data are the mean ± SD).

### 3.2. Subject Adherence

A total of 25 subjects completed all four interventions, one subject completed three, and four subjects completed two (Figure 1). Overall data from 30 subjects were included for BP and microcirculation measures, 28 subjects for urinary excretion measures, and 20 subjects were included in fecal excretion, and mineral retention analyses, for whom these measures were available. Three subjects were on some form of hypertensive monotherapy and were monitored to ensure timing and dose of medication did not change during the study duration. The mean body weight change for the control, potato, FF, and supplement interventions was −1.1 kg, −0.9 kg, −0.9 kg, and −1.0 kg, respectively. Supplement intake compliance by pill count was 91%. Overall dietary intake compliance was 98–99% for all interventions. Potassium intake compliance for control, potato, FF, and supplement was 98, 98, 99, and 96%, respectively.

### 3.3. Chemical Analysis of Controlled Diets

Potassium content for the intervention phases of control, potato, Frenchf (FF), and supplement were 57.4 ± 1.2 mmol/d (2238 ± 44 mg/d), 77.1 ± 0.1 mmol/d (3008 ± 4), 76.3 ± 0.8 mmol/d (2978 ± 29, and 84.6 ± 1.1 mmol/d (3299 ± 44 mg/d), respectively. These levels were within 9% of the 84.6 mmol/d (3300 mg/d) target. For Na, the target for all four interventions was set at 143.5 mmol/d (3300 mg/d), analysis showed Na content to range between 90 and 103% of the target.

### 3.4. Blood Pressure

The time course of blood pressure throughout each intervention is shown in Figure 2. Primary BP outcome analysis was a comparison of end-of-treatment SBP and DBP for all interventions, with an adjustment for baseline BP. There were no significant differences among interventions (all values mmHg; control: 129.3 ± 0.90/83.7 ± 0.74, potatoes: 126.2 ± 0.93/83.8 ± 0.76, French fries: 127.8 ± 0.95/83.6 ± 0.77, supplement: 128.5 ± 0.90/84.1 ± 0.74; SBP: *p* = 0.08, DBP: *p* = 0.9). However, differences for SBP between control and potato phases approached significance (*p* = 0.06; Figure 3). Secondary BP outcome analysis assessed change in BP over time. For the change in BP over the course of each intervention, there were also no significant differences among interventions (all values mmHg; control: −2.6 ± 0.9/−1.2 ± 0.6, potatoes: −6.0 ± 1.1/−1.3 ± 0.7, French fries: −4.2 ± 1.0/−1.5 ± 0.9, supplement: −3.4 ± 0.9/−0.8 ± 0.8; SBP: *p* = 0.07, DBP: *p* = 0.9). However, the potato intervention resulted in a SBP change of -6.01 mmHg, which, after contrast analysis, was significantly lower (*p* = 0.011) when compared with the SBP change in the control diet (−2.6 mmHg; Figure 4). There were no significant differences comparing men and women for any BP outcomes. There were no significant differences in endothelial function among treatments (all values ≥ 87% CVC max, *p* > 0.05).

### 3.5. Urinary Potassium and Sodium Excretion

There were significant differences among groups in both urinary potassium (*p* < 0.0001) and Na (*p* < 0.0001) excretion (Figure 5). For potassium, the control phase urinary excretion (38.4 ± 2.6 mmol/d or 1504 ± 100 mg/d) was significantly lower than the potato (49.3 ± 3.9 mmol/d or 1926 ± 152 mg/d; *p* < 0.0001), FF (46.4 ± 3.1 mmol/d or 1816 ± 120 mg/d; *p* = 0.004), and supplement (50.9 ± 4.1 mmol/d or 1989 ± 162 mg/d; *p* < 0.0001) phases. For Na, the potato phase (121.3 ± 7.3 mmol/d or 2791 ± 168 mg/d) was significantly higher than both the control (94.9 ± 5.3 mmol/d or 2183 ± 121 mg/d; *p* < 0.0001) and supplement (91.1 ± 5.5 mmol/d or 2095 ± 127 mg/d; *p* < 0.0001) interventions, and was also significantly higher with the FF intervention (110.5 ± 6.1 mmol/d or 2524 ± 141 mg/d) compared to the control (*p* = 0.02) and supplement (*p* = 0.0007) phases.

### 3.6. Potassium and Sodium Fecal Excretion, Balance, and % Absorption

There were no significant differences (*p* = 0.29) in fecal potassium excretion among interventions (Figure 5). Differences in potassium and Na retention were assessed as absolute and % of intake. Absolute potassium retention was significantly higher during the supplement (28 ± 4.6 mmol/d or 1093 ± 178 mg/d) compared to the control (14.5 ± 3.2 mmol/d or 568 ± 125 mg/d; *p* < 0.0001), potato (20.6 ± 3.9 mmol/d or 807 ± 154 mg/d; *p* = 0.04), and FF (21.2 ± 3.5 mmol/d or 829 ± 138 mg/d; *p* = 0.07) study phases. However, when assessed as percent of intake (%), there were no significant differences (*p* = 0.09) between interventions (Figure 6).

There were no significant differences (*p* = 0.11) in fecal Na excretion. Absolute Na retention was significantly lower during the potato (21.4 ± 8.6 mmol/d or 492 ± 197 mg/d) compared to the control (38.6 ± 7 mmol/d or 888 ± 161 mg/d; *p* = 0.02), FF (38.2 ± 6.7 mmol/d or 877 ± 153 mg/d; *p* = 0.03), and supplement (44.2 ± 6.7 mmol/d or 1017 ± 153 mg/d; *p* = 0.0013) study phases. Sodium retention in the potato group, when assessed as % of intake (12.8 ± 6.0), remained significantly lower than control (27.0 ± 5.2; *p* = 0.007), FF (24.1 ± 4.6; *p* = 04) and supplement (30.9 ± 4.8; *p* = 0.0003) phases (Figure 7).

There were significant differences in percent net absorption (%ab) for both potassium (*p* = 0.005) and Na (*p* = 0.025). For potassium, the %ab in the supplement phase (91.9 ± 1.0) was significantly higher compared to the control (88.3 ± 1.3; *p* = 0.003). For Na, %ab was significantly lower in the potato intervention (96.2 ± 0.5) compared to the FF (97.0 ± 0.4; *p* = 0.047) and supplement (97.1 ± 0.4; *p* = 0.03) interventions.

## 4. Discussion

This study is the first controlled feeding trial looking specifically at dietary potassium as the primary nutrient of interest on effects on BP in vulnerable men and women. We assessed the effect of dietary potassium from potato sources (baked and boiled (potato) or French fry (FF)) compared with a potassium supplement (K gluconate) on BP and metabolism outcomes in pre-hypertensive to hypertensive adults via a cross-over, highly controlled feeding trial. Potatoes comprise ~20% of the vegetable intake in the American diet, with white potatoes and French fries representing ~7 and 3% of overall potassium intake [2]. End-of-treatment BP comparisons were selected as the primary outcome measure because of errors associated with high screening and baseline BP, attributed to white coat syndrome, or similar participant unsettling due to unfamiliar circumstances on initial BP measurements [37]. While there were no significant differences among treatments for either end of intervention SBP or DBP outcomes, increased potassium from baked or boiled potatoes approached a significant benefit to SBP compared with the control group by 3.1 mm Hg (*p* = 0.06). Moreover, there was a significant change from baseline in SBP between the potato (−6.01 mm Hg) compared with the control diet (−2.6 mm Hg) (*p* = 0.011). Such a large change in BP over time is clinically relevant. 

Current evidence on the effect of dietary potassium as a specific nutrient of interest on BP only exists from interventions utilizing dietary advice with mixed results [13,14,15]. In an earlier study, Chalmers et al. assessed the effects of both increasing dietary potassium and reducing dietary sodium (Na), via nutrition coaching, on BP in hypertensive individuals from an Australian cohort [13]. After 12 weeks, subjects (age 52.3 ± 0.8 years; 181 males and 31 females) placed on the high potassium intake diet (>100 mmol/3900 mg potassium/d) had significant decreases in both SBP and DBP of 7.7 ± 1.1 and 4.7 ± 0.7 mm Hg, respectively. However, the other invention arms (low-Na arm, low-Na/high-potassium arm) had similar reductions in BP, highlighting the possibility that the effect was due to overall diet change, and not a specific nutrient (e.g., potassium). Findings were similar in another Australian cross-over study utilizing dietary advice and supplementation to examine the effects of high vs. low Na on BP in the framework of a high-potassium diet (~87 mmol/d or 3400 mg/d) [30]. SBP was reduced by 5.5 mm Hg with potassium supplementation despite high Na intakes of supplementation up to 120 mmol/d or 2760 mg/d. However, the study lacked a control arm, making it difficult to conclude whether the effects were due specifically to potassium intake. More recent studies utilizing dietary advice to manipulate dietary potassium had returned null results [14,15]. In a study conducted in the United Kingdom, investigators assessed the effects of increased potassium intake from both dietary sources and supplements on BP in pre-hypertensive individuals (n = 48, 22–65 years) [14]. In a cross-over study, subjects completed four, 6 week dietary interventions including a control diet, an additional 20 or 40 mmol potassium/d (780 or 1560 mg/d) from fruit and vegetables, and 40 mmol potassium citrate/d, with no significant differences in BP among intervention periods. Several design differences could explain discrepant results. Compared with the Chalmers study, subjects had less of an increase in dietary potassium (20–40 mmol/d or 780–1560 mg/d increase compared with 100 mmol or 3900 mg/d) as well as lower baseline BP (SBP: 137.7 vs. > 150 mmHg, DBP: 88.6 vs. > 95 mmHg). Coaching participants to increase dietary potassium from fruit and vegetable sources with or without Na intake advice [13] is less effective than using a controlled diet. However, Miller et al. [15] found similar results from an effectiveness study in an urban African American population [15].

Approximately 60–70% of the thermal hyperemic vasodilatory response is mediated by the production and release of NO [38,39]. Initial changes in the microvasculature, as measured in the capillaries and arterioles of the skin, may be causally related to endothelial dysfunction, considered the initiating event in the development of CVD and related mortalities. The lack of significant differences in endothelial function due to dietary interventions in our study may have been due to the approach we used [40]. The temperature used in our protocol to elicit a maximal vasodilatory response was only one degree Celsius above the plateau phase (indirect measure of NO release and endothelial function). This may have produced a ceiling effect in which the plateau phase was already causing maximal vasodilation, making it difficult to determine any treatment effects. The decrease in perfusion pressure for each intervention (especially during the first week) also may have blunted any possible vasodilatory benefit due to dietary potassium.

The primary site of regulation for both potassium and Na is the kidney. Potassium urinary excretion averaged 63% and Na excretion averaged 75% of total intake across all interventions, similar to values that have been reported in other controlled diet studies [16,41]. Potassium as a supplemental salt was retained the most efficiently, reflecting higher net absorption efficiency. Of the seven trials examined by the 2019 DRI committee, potassium intake level of approximately 56 mmol/d or 2200 mg/d or greater resulted in a positive balance [20,21,22,23,24,42] levels of 54 mmol/d or 21 mg/d or less resulted in a negative balance [22,24], and one study reported neutral balance at an intake of 52 mmol/d or 2034 mg/d [43]. Our study evaluated potassium intake levels above 59 mmol/d or 2300 mg/d and resulted in positive potassium balance in all phases. 

Despite the greater retention of potassium from the salt, the greatest benefit on sodium retention reduction was from the potato intervention. The greater reduction in sodium retention with potatoes was associated with greater BP reduction (SBP) compared to the control diet. The benefit of the potato intervention at reducing Na retention was even more impressive considering the Na intake in the potato intervention was higher by 17 mmol/d (400 mg/d) compared to both the control and supplement intervention.

The influence of dose and form are important factors in potassium delivery in the context of influencing Na retention. Sodium and water reabsorption in the kidney are tightly linked, and the role of potassium within this relationship and overall fluid balance is thought to be important, but not well understood. In animal models, both potassium depletion and potassium loading can lead to an increase via Na/H+ exchanger upregulation in the proximal kidney, and decrease, via Na-chloride cotransporter deactivation in the distal kidney, in Na reabsorption [44,45,46]. Differences in Na retention may have also been the result of the non-mineral component of the potassium supplement, gluconate. Gluconic acid is an oxidized product of glucose, and active Na absorption in the upper small intestine occurs through the Na+/glucose cotransporter, a process largely dependent on the presence of glucose. This may explain the higher % absorption of Na in the K gluconate group (97.1%), and the difference in Na retention, although the amount (~460 mg/90 mg of potassium (per pill)) of gluconate administered may not have been enough to cause this effect.

Strengths of our study include the highly controlled diet and cross-over design to compare potassium supplementation from food and salts on BP and CVD risk factors in an unhealthy population with excellent compliance. The bulk of the evidence to date on the relationship of dietary potassium and health use self-assessment of dietary intake or biomarkers of intake such as 24 h urine or spot urines. In addition to the inherent error associated with these assessment methods, there is also a large inter-individual variability with excretion measures, especially when only represented by one time point. Limitations include small sample size, a cohort of convenience, poor retention, and a relatively short duration. Intervention duration in the potassium supplementation trials reviewed by the AHRQ ranged from 4 weeks to 36 months, but the longest dietary interventions were 8–12 weeks. However, a recent re-analysis by the DASH group showed diet-induced changes in BP were achieved within 1 week on a controlled DASH dietary pattern [47].

This was the first controlled feeding study of potassium from food and supplements on BP and CVD outcomes in an unhealthy population. This small trial can inform the design of future studies to determine dose–response effects of potassium on BP and CVD risk factors and potassium and sodium retention. Of public health relevance is our observation that French fries in amounts typical of a large serving in a fast food restaurant has no adverse effect on blood pressure or endothelial function.

## Figures and Tables

**Figure 1 nutrients-13-01610-f001:**
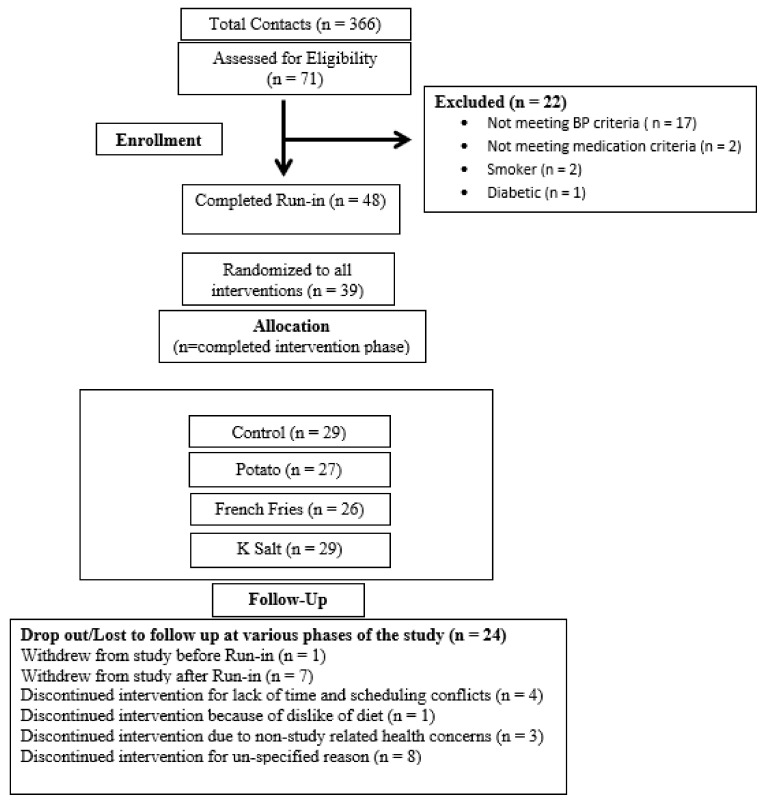
Flow Diagram of Potassium Hypertension and Retention Study.

**Figure 2 nutrients-13-01610-f002:**
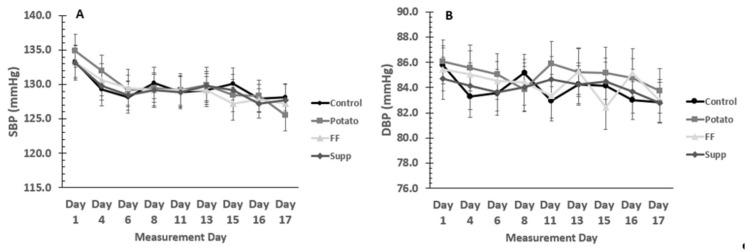
Blood pressure throughout each intervention. (**A**) Systolic blood pressure (SBP) and (**B**) diastolic blood pressure (DBP) means (N = 30) for each measurement day in each intervention phase.

**Figure 3 nutrients-13-01610-f003:**
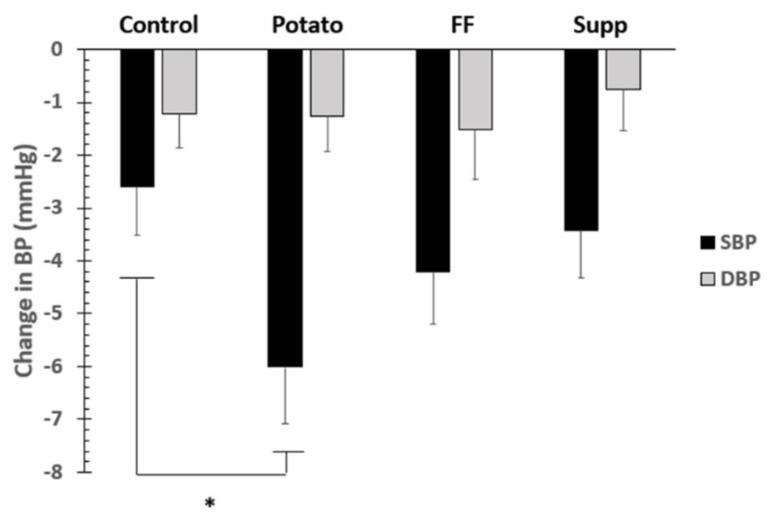
Secondary BP outcome: change in BP over time. * denotes significant difference (*p* < 0.05) between potato and other groups.

**Figure 4 nutrients-13-01610-f004:**
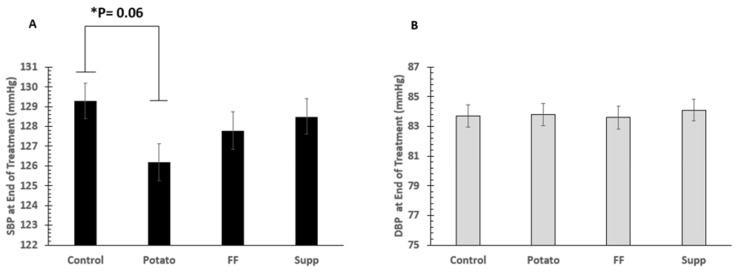
Primary BP outcome: comparison of end-of-treatment (**A**) systolic blood pressure (SBP) and (**B**) diastolic blood pressure (DBP) for all intervention phases with an adjustment for baseline BP.

**Figure 5 nutrients-13-01610-f005:**
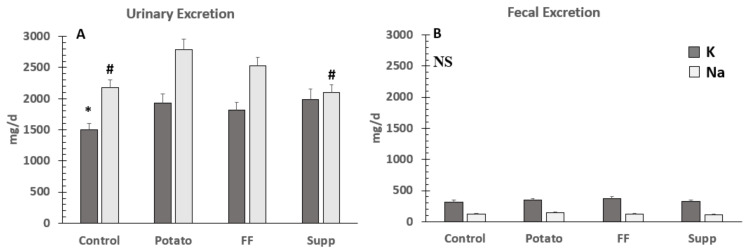
(**A**) Urinary (N = 28) and (**B**) Fecal (N = 20) excretion of potassium and sodium in hypertensive adults fed a typical potassium intake (control) and supplemented potassium through food (baked or boiled potatoes or French fries (FF)) or supplement. * denotes significant differences between control and potassium groups (*p* < 0.05) for potassium retention. # denotes significant differences between groups for sodium retention (i.e., control and supplement were both significantly different from potato and FF, but not each other). NS, no significant differences. To convert to mmol/d for potassium, divide by 39; and for sodium, divide by 23.

**Figure 6 nutrients-13-01610-f006:**
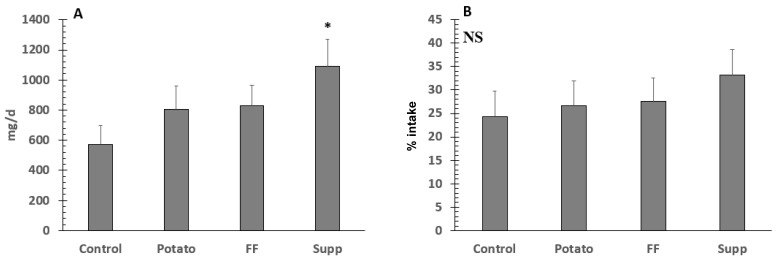
Potassium retention as mg/d (**A**) and percent (%) of intake (**B**) in hypertensive adults on a controlled diet. * denotes significant difference (*p* < 0.05) between supplement and other groups. NS, not significant differences. To convert to mmol/d, divide by 39.

**Figure 7 nutrients-13-01610-f007:**
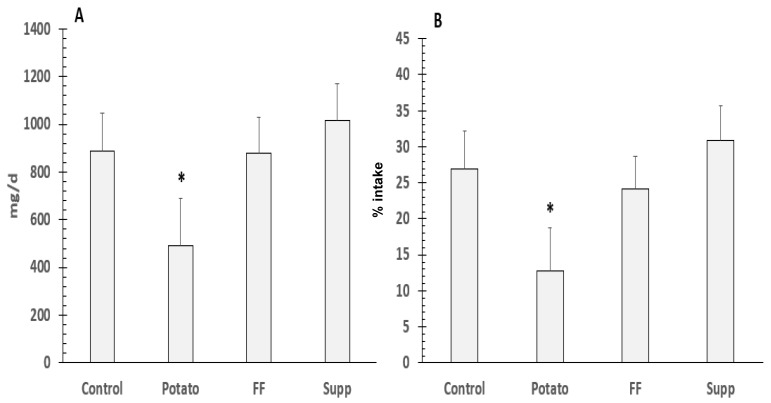
Differences in sodium retention as mg/d (**A**) and percent (%) of intake (**B**) in hypertensives on a controlled diet. * denotes significant difference (*p* < 0.05) between potato and other groups. To convert to mmol/d, divide by 23.

**Table 1 nutrients-13-01610-t001:** Baseline characteristics of all participants who completed at least two intervention periods.

	All Mean (SD)	Male Mean (SD)	Female Mean (SD)
n	30	15	15
Age (year)	48.2 (15.0)	43.8 (13.7)	52.7 (15.4)
Height (cm)	172.2 (10.2)	179.4 (7.1)	165.0 (7.2)
Weight (kg)	93.9 (22.9)	99.1 (20.7)	88.6 (24.4)
BMI (kg/m^2^)	31.4 (6.1)	30.5 (4.8)	32.3 (7.2)
Systolic Blood Pressure (SBP) (mmHg)	133.6 (12.2)	133.8 (13.8)	133.3 (10.4)
Diastolic Blood Pressure (DBP) (mmHg)	85.5 (8.6)	86.1 (8.1)	84.9 (9.2)

## Data Availability

All data are contained in the article or available upon reasonable request.

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
