# Peer review of "Short-Term RCT of Increased Dietary Potassium from Potato or Potassium Gluconate: Effect on Blood Pressure, Microcirculation, and Potassium and Sodium Retention in Pre-Hypertensive-to-Hypertensive Adults"

_nutrients, 2021, doi:10.3390/nu13051610_

Round 1
Reviewer 1 Report
Stone et al. investigated the effect of different 17-day 26-mmol potassium supplements on the micro and macrocriculation in subjects with (pre-)hypertension). The authors have performed an extensive potassium balance study. The authors found SBP was lower after supplementation with potatoes while total potassium retention was highest with supplement.
Major
- The authors should consider to first test the effect of the control diet versus the averaged data of the different potassium supplementations and thereafter test whether any difference is seen among the dietary interventions. This approach is more in line with their primary research question: is there any difference in the micro- and macrocirculatory effects of different potassium supplementation strategies. I guess that this approach may change the BP conclusion to: potassium supplementation significantly decreases BP but no significant differences are seen between different potassium supplementation strategies.
- The authors have not described their data on endothelial function in the results section
- The potassium intake of all diets is below the WHO advised intake of 90 mmol/day
Minor
- Was antihypertensive treatment stable in the 6 weeks prior to the study?
- Why did the authors use baseline BP (run in day 1-4) as a covariate instead of the control diet. In contrast to the control diet, the dietary potassium intake is not known during the run in day 1-4)
- Figure 5: a and b denote significant differences compared to what?
- The authors should not make too much use describing trends such as ‘trended towards significantly different’. By providing absolute measures and p-value the readers are well informed about the magnitude of the changes
- The authors could consider measuring the potassium balance difference for the entire study (instead of a daily balance) diet as weekly rhythmic changes in potassium excretion have been observed. How many mmol was retained per diet?
- How do the authors explain the potassium retention during control diet?
- Did the amount of sodium and/or potassium retention correlate with the blood pressure reduction?
- The authors discuss that the BP reduction after potatoes (6.1 mmHg) was larger than control treatment (2.1 mmHg). Did the authors correct for baseline BP in this analysis?
Author Response
Reply: We believe the statistical strategies effectively assess our main research question of whether different forms of potassium have differential effects on both blood pressure and retention. Combining all increased potassium arms of the study would take away from the primary finding of the potato form having a potential effect compared to all other study arms and the purpose of our original research design. This is stated in lines 312 – 316, however lines 87 – 88 in the introduction have been modified slightly to ensure this message is properly conveyed.
- The authors have not described their data on endothelial function in the results section
Reply: This is now addressed in lines 256 – 258.
- The potassium intake of all diets is below the WHO advised intake of 90 mmol/day
Reply: The control diet was intended to be well below public health recommendations. While slightly lower than the WHO guidelines, the other diets were still relatively close to these recommendations (~3500mg/d vs. 3000 – 3300mg/d), and were also more in line with the current NASEM DRI potassium Adequate Intake of 3000mg/d for healthy adult men and women. Further, reports by both the WHO and DRI committee make it clear that a sufficient intake of potassium for healthy individuals (let alone a vunerable population such as those who are hypertensive) has yet to be determine due to insufficient research.
Minor
- Was antihypertensive treatment stable in the 6 weeks prior to the study?
Reply: We only assessed if participants were currently taking antihypertensive medication as described in lines 226 -228.
- Why did the authors use baseline BP (run in day 1-4) as a covariate instead of the control diet. In contrast to the control diet, the dietary potassium intake is not known during the run in day 1-4)
Reply: We used baseline BP at the beginning of each phase, as blood pressure is a highly variable biomarker. We determined that assessing it this way would most accurately characterize any true changes assessed at end of treatment for each phase (e.g., BP was higher at the beginning of one phase compared to another phase so we needed to control for this). The control diet was its’ own research arm specifically designed with a lower potassium intake, making BP values taken during this study period only applicable to the control arm of the study.
- Figure 5: a and b denote significant differences compared to what?
Reply: Different letters denote significant differences between potassium source and control as now described.
- The authors should not make too much use describing trends such as ‘trended towards significantly different’. By providing absolute measures and p-value the readers are well informed about the magnitude of the changes
Reply: All references to “trends” have been removed (lines 284 – 288, now 289-293).
- The authors could consider measuring the potassium balance difference for the entire study (instead of a daily balance) diet as weekly rhythmic changes in potassium excretion have been observed. How many mmol was retained per diet?
Reply: We included daily balance as this is the most common way balance/retention data are presented in the extant literature and we want this research to be as widely comparable and translatable as possible. Furthermore, the effort was made to determine daily balances to be able to compute the variance. If one wishes to know the respective mineral retention for the entire study duration they would only need to multiple the average by the number of observation days (e.g., potassium daily average x 13 days).
- How do the authors explain the potassium retention during control diet?
Reply: This is addressed in the discussion (lines 372 – 381). As discussed, positive potassium balance has been observed by others at intake levels below what was administered in the control diet.
- Did the amount of sodium and/or potassium retention correlate with the blood pressure reduction?
Reply: Correlations were not significant, but we were underpowered to assess bivariate associations.
- The authors discuss that the BP reduction after potatoes (6.1 mmHg) was larger than control treatment (2.1 mmHg). Did the authors correct for baseline BP in this analysis?
Reply: This measurement inherently controls for baseline BP. Change in BP is assessed as end of treatment – baseline BP. This is stated in lines 152 – 154.
Reviewer 2 Report
Overall, although this is interesting short-term RCT, there are several important points that need to be addressed.
We know that kidney function/eGFR are important variables and affect the blood pressure and sodium retention in pre-hypertensive-to-hypertensive adults.
Thus, the strongly advise the investigators to include baseline eGFR of study participants in the results and also in Tables.
The details on Randomization and study designs can be improved.
Is this prospective Randomized blinded?
This study has not been registered in ClinicalTrials.gov?
What method is being used to ensure allocation concealment?
Author Response
Reply: We did not assess eGFR directly, however we did screen for renal disease/disorders and excluded any individuals to this end. This is stated in the exclusion criteria (line 99).
Thus, the strongly advise the investigators to include baseline eGFR of study participants in the results and also in Tables.
Reply: The cross-over design and short duration of the study render between subject differences in eGRF and other variables unlikely to alter the results of the intervention. Moreover, it has been shown that elevated blood pressure alone does not significantly impact GFR decline (Eriksen et al. 2016doi: 10.1016/j.kint.2016.03.021, Eriksen et al. 2017, doi: 10.1186/s12882-017-0496-7) and because our participants were screened for other comorbidities it is unlikely that GFR change had any impact on our results. Given the duration of the study (4-6 months) it is also unlikely that a significant GFR decline would have occurred in this short time frame.
The details on Randomization and study designs can be improved.
Reply: The randomization was done by a computer-generated allocation (added to lines 108 -109).
Is this prospective Randomized blinded?
Reply: This study was not blinded. In a controlled feeding study, blinding is usually not possible as the participants know what food they are eating when they eat it (e.g., if they were on the French fry arm there was no way to blind them to receiving the French fries).
This study has not been registered in ClinicalTrials.gov?
Reply: This study is registered on ClinicalTrials.gov; this information is on line 22.
What method is being used to ensure allocation concealment?
Reply: Allocation concealment was not possible due to the reasons stated above. This is why we do not refer to the study as being blinded in any way throughout the paper.
Round 2
Reviewer 1 Report
The authors should change the way how they describe significant results in their figures. Is b significantly different from all other b's? This seems nog the case. It is unclear what the difference between a and b is.
Author Response
Figure captions were changed to more clearly distinguish significant differences.
Reviewer 2 Report
I reviewed the revised manuscript and the response to reviewers' comments. Revised Manuscript is well written. All comments have been addressed .
Author Response
Thank you